# Vitamin D Receptor Antagonist MeTC7 Inhibits PD-L1

**DOI:** 10.3390/cancers15133432

**Published:** 2023-06-30

**Authors:** Negar Khazan, Emily R. Quarato, Niloy A. Singh, Cameron W. A. Snyder, Taylor Moore, John P. Miller, Masato Yasui, Yuki Teramoto, Takuro Goto, Sabeeha Reshi, Jennifer Hong, Naixin Zhang, Diya Pandey, Priyanka Srivastava, Alexandra Morell, Hiroki Kawano, Yuko Kawano, Thomas Conley, Deepak M. Sahasrabudhe, Naohiro Yano, Hiroshi Miyamoto, Omar Aljitawi, Jane Liesveld, Michael W. Becker, Laura M. Calvi, Alexander S. Zhovmer, Erdem D. Tabdanov, Nikolay V. Dokholyan, David C. Linehan, Jeanne N. Hansen, Scott A. Gerber, Ashoke Sharon, Manoj K. Khera, Peter W. Jurutka, Natacha Rochel, Kyu Kwang Kim, Rachael B. Rowswell-Turner, Rakesh K. Singh, Richard G. Moore

**Affiliations:** 1Wilmot Cancer Institute and Division of Gynecologic Oncology, Department of Obstetrics and Gynecology, University of Rochester Medical Center, Rochester, NY 14642, USAalexandra_morell@urmc.rochester.edu (A.M.); kyukwang_kim@urmc.rochester.edu (K.K.K.); richard_moore@urmc.rochester.edu (R.G.M.); 2Department of Environmental Medicine, University of Rochester Medical Center, Rochester, NY 14642, USA; 3Department of Microbiology and Immunology, University of Rochester Medical Center, Rochester, NY 14642, USA; 4Department of Pathology and Laboratory Medicine, University of Rochester Medical Center, Rochester, NY 14642, USA; yabomabo@gmail.com (M.Y.);; 5School of Mathematical and Natural Sciences, University of Arizona College of Medicine, Phoenix, AZ 85004, USA; 6Department of Medicine, Hematology/Oncology, University of Rochester Medical Center, Rochester, NY 14642, USAthomas_conely@urmc.rochester.edu (T.C.);; 7Division of Surgical Research, Rhode Island Hospital, Brown University, Providence, RI 02912, USA; naohiro_yano@brown.edu; 8Center for Biologics Evaluation and Research, U.S. Food and Drug Administration, Silver Spring, MD 20993, USA; 9CytoMechanobiology Laboratory, Department of Pharmacology, Penn State College of Medicine, Pennsylvania State University, Hershey, PA 17033, USA; 10Department of Pharmacology, Department of Biochemistry & Molecular Biology, Center for Translational Systems Research, Penn State College of Medicine, Pennsylvania State University, Hershey, PA 17033, USA; nxd338@psu.edu; 11Division of Surgery, University of Rochester Medical Center, Rochester, NY 14642, USA; 12Department of Psychological and Brain Sciences, Colgate University, Hamilton, NY 13346, USA; 13Division of Surgery and Microbiology and Immunology, University of Rochester Medical Center, Rochester, NY 14642, USA; 14Department of Radiation Oncology, University of Rochester Medical Center, Rochester, NY 14642, USA; 15Birla Institute of Technology, Ranchi 835215, India; 16Presude Lifesciences, Delhi 110075, India; 17School of Mathematical and Natural Sciences, Arizona State University, Health Futures Center, Phoenix, AZ 85054, USA; 18Institute of Genetics and of Molecular and Cellular Biology, 67400 Illkirch-Graffenstaden, France

**Keywords:** PD-L1, AML, MDS, vitamin D, VDR, small molecule, M2H assay, efferocytosis

## Abstract

**Simple Summary:**

Programmed death-ligand 1 (PD-L1) enables immune evasion of tumors. Antibodies targeting PD-L1/PD-1 exhibit durable responses in eligible patients. However, antibodies cause life-threatening toxicities. Small molecules targeting PD-L1, or the drivers of PD-L1 or PD-L1/PD-1 axis, are being explored as alternatives. Thus, identifying vitamin D/vitamin D receptor (VDR) as the driver of PD-L1 expression in AML significantly enhances our understanding of the origin of PD-L1-driven immune evasions in AML and malignancies of pancreas and ovaries, where similar transcriptional regulation has been observed. To target vitamin D/VDR, we have developed MeTC7, which inhibits PD-L1 expression in vitro and in vivo and provides a new approach to block PD-L1/PD-1-driven tumorigenesis.

**Abstract:**

Small-molecule inhibitors of PD-L1 are postulated to control immune evasion in tumors similar to antibodies that target the PD-L1/PD-1 immune checkpoint axis. However, the identity of targetable PD-L1 inducers is required to develop small-molecule PD-L1 inhibitors. In this study, using chromatin immunoprecipitation (ChIP) assay and siRNA, we demonstrate that vitamin D/VDR regulates PD-L1 expression in acute myeloid leukemia (AML) and myelodysplastic syndrome (MDS) cells. We have examined whether a VDR antagonist, MeTC7, can inhibit PD-L1. To ensure that MeTC7 inhibits VDR/PD-L1 without off-target effects, we examined competitive inhibition of VDR by MeTC7, utilizing ligand-dependent dimerization of VDR-RXR, RXR-RXR, and VDR-coactivators in a mammalian 2-hybrid (M2H) assay. MeTC7 inhibits VDR selectively, suppresses PD-L1 expression sparing PD-L2, and inhibits the cell viability, clonogenicity, and xenograft growth of AML cells. MeTC7 blocks AML/mesenchymal stem cells (MSCs) adhesion and increases the efferocytotic efficiency of THP-1 AML cells. Additionally, utilizing a syngeneic colorectal cancer model in which VDR/PD-L1 co-upregulation occurs in vivo under radiation therapy (RT), MeTC7 inhibits PD-L1 and enhances intra-tumoral CD8^+^T cells expressing lymphoid activation antigen-CD69. Taken together, MeTC7 is a promising small-molecule inhibitor of PD-L1 with clinical potential.

## 1. Introduction

The programmed death-ligand-1 (PD-L1)/programmed cell death protein-1 (PD-1) axis is a well-characterized immune evasion mechanism of tumors [1,2]. Several other checkpoint molecules, such as CTLA-4 [3] and V-domain Ig suppressor of T-cell activation (VISTA) [4], are also involved in the immune escape of tumors. Drivers of PD-L1 expression on tumor cells have been identified [5,6]. Calcitriol, the active form of vitamin D, was shown to upregulate PD-L1 in AML and head/neck cancer cell lines [7]. Phorbol esters, retinoic acid, and interferon-gamma also regulate PD-L1 expression in tumor cells [8,9,10].

Monoclonal antibodies targeting the PD-L1/PD-1 axis have shown lasting responses in melanoma and carcinomas of the breast, bladder, cervix, uterus, and lung [11]. A portion of cancer patients do not benefit from PD-L1/PD-1 targeted immunotherapies [12]. In addition, monoclonal antibodies cause life-threatening immune adverse reactions, face resistance, and carry poor pharmacodynamics, e.g., the inability to penetrate tumors and inhibit intracellular or exosomal PD-L1 [13]. The intracellular reservoir of PD-L1 can rebound to cell membranes and reduce immunotherapy outcomes [14]. Similarly, exosomal PD-L1 can promote immune tolerance and negate immunotherapy outcomes [15,16]. Further, secreted splice variants of PD-L1 can entrap PD-L1 targets in antibody therapies limiting their effectiveness [17].

There is a growing interest in the development of small molecules that can inhibit PD-L1 or block PD-L1/PD-1 interactions similar to antibodies [18,19,20,21,22]. Examples of the leading small-molecule inhibitors of the PD-L1/PD-1 axis include BMS202, CA-170, and INCB086550 [18,19,20,21,22,23]. Identifying small-molecule inhibitors of PD-L1 is impaired by: (1) the flat topology of PD-L1, which impairs small molecules’ ability to dock, and (2) the lack of the identity of drug-responsive regulators of PD-L1 [10]. We and Dimitrov et al. [7] have identified vitamin D/VDR as a drug-responsive driver of PD-L1 in AML, MDS, and ovarian and pancreatic cancer cells. Targeting VDR is challenged by the unavailability of pharmacologically pure antagonists. Recently, we described the identification of MeTC7, a pharmacologically pure VDR antagonist [24].

In this study, we aim to investigate the mechanism of VDR inhibition using a mammalian 2-hybrid (M2H) assay to validate its actions against VDR and dimerization of VDR-RXR, an obligatory step in the activation of VDR signaling. Furthermore, we aim to examine whether antagonist binding also blocks the formation of VDR complexes with essential coactivators (SRC, DRIP205) needed for gene inductions by VDR. Next, we aimed to examine whether MeTC7 can inhibit PD-L1 expression in a panel of diverse tumor cell types (AML, ovarian and pancreatic cancer) in vitro and in a syngeneic model that converges overexpression of both VDR/PD-L1 under the same settings. Taken together, this study deepens our understanding of the pharmacologic mechanism of actions of MeTC7 and presents it as a promising small-molecule inhibitor of PD-L1.

## 2. Materials and Methods

Study design: VDR overexpression in AML subtypes and its impact on survival will be determined by analysis of AMPL patients’ microarray databases (Bloodspot and R2-Genomics and Visualization platform). The association of vitamin D/VDR with PD-L1 will be validated by a CHIP assay and siRNA knockdown of VDR. The mechanism of VDR antagonism of MeTC7 will be examined by M2H assay. PD-L1 expression in MeTC7-treated primary AML or cell lines will be examined by immunoblotting. The effects of MeTC7 on the viability and clonogenicity of AML will be examined by MTS assay, colony counts, and microscopy. The adhesion of AML cells with mesenchymal stem cells will be examined by co-culture experiments. The effect of MeTC7 on efferocytosis will be examined by co-culture of THP-1 cells with apoptosed neutrophils. In vivo anti-PD-L1 activity will be examined by a syngeneic mouse colon cancer model.

Collection of untreated and relapsed AML cells: Primary AML, relapsed AML, and bone marrow (BM) cells were isolated from AML patients and healthy donors who had provided their informed consent at the University of Rochester Medical Center.

VDR expression in human AML and correlation with survival: VDR mRNA expression in AML subtypes versus normal hematopoietic stem cells (HSC) controls were established via analysis of the Bloodspot database of AML patients comparing to normal hematopoiesis (https://servers.binf.ku.dk/bloodspot/?gene=VDR&dataset=normal_human_v2_with_AMLs (accessed on 26 June 2023)). The association of VDR mRNA overexpression with survival in AML patients was analyzed using the R2-Genomics.org platform (Database: Bohlander_422-Mas5.0-u13a; https://hgserver1.amc.nl/cgi-bin/r2/main.cgi (accessed on 26 June 2023)). The system-selected VDR mRNA expression cut-off was set to 70.8. The expression of VDR and PD-L1 proteins in cultured AML primary cells and cell lines was analyzed using Western blot, as described previously [24].

Cell lines and MeTC7: AML cell lines (MOLM13, MV411, THP1, and U937) purchased from American Type Culture Collection (ATCC, Manassas, VA) were cultured in RPMI medium (Gibco, cat #22400) supplemented with 10% fetal bovine serum (FBS) (Hyclone, #SH30396.03) and 1% pen-strep (Gibco, cat#15140122).

Chromatin Immunoprecipitation (ChIP): ChIP assay was performed using the Magna ChIP kit (Millipore; #17-10085) according to the manufacturer’s instructions. The detailed method of ChIP assay is described in the Appendix A section.

siRNA knockdown of VDR: Stable VDR knockdown in AML cell lines was carried out using siRNA. Methods for siRNA knockdown are described in the Appendix A section.

Mammalian-2-hybrid (M2H) assay: Human embryonic kidney cells (HEK293) purchased from ATCC were used to perform the mammalian-2-hybrid assays (M2H). The cells were seeded in a 24-well plate at a density of 70,000 cells/well in DMEM (Hyclone, Logan, UT, USA) supplemented with 10% FBS (Atlanta Biologicals, Flowery Branch, GA, USA), 100 µg/mL streptomycin, 100 U/mL penicillin (Caisson labs, Smithfield, UT, USA) 22–24 h prior to transfection were transiently transfected using Polyethylenimine (PEI) (Santa Cruz Biotechnology, Dallas, TX, USA) according to the manufacturer’s protocol. The cells in each well received 20 ng of bait vectors and prey vectors, 250 ng of the luciferase reporter plasmid (pFR-luc), and 20 ng of pRL-null renilla control plasmid along with 1.25 µL of PEI reagent. After 22–24 h of transfection, the cells were treated with reference compound (i.e., 1,25D) either alone or in combination with MeTC7. For the RXR-RXR assay, the cells were treated with reference compound (i.e., bexarotene) and/or MeTC7. All compounds were solubilized in ethanol except MeTC7, which was solubilized in DMSO. After 24 h of treatment, the cells were harvested into 5X cell lysis buffer, and the whole cell lysates were collected and quantified for transcriptional activity using the Dual Luciferase Assay System (Promega, Madison, WI, USA) in a Sirius FB12 luminometer (Berthold Detection Systems, Pforzheim, Germany).

MTS assay, Western blot, and colony formation assays: Cell viabilities and Western blot analyses were performed as published previously [24]. Details of the antibodies (source and catalog numbers) are described in the Materials and Methods section. Methods for estimation of colony formation in control versus MeTC7 cells are described in detail in the Appendix A section. Image J software (ImageJ 1.53 t, Java 1.8.0_322 (64-bit) was employed for the densitometric analysis of Western blot images. Original scans of the Western blot images described in the report are provided in Appendix A.

Co-culture of mesenchymal stem cells (MSCs) with primary cells: MSCs were isolated from the low-density bone marrow and were co-cultured with AML primary cells in 8-well glass chamber slides following the methods published previously [25]. Details of the isolation and co-culture are described in the Appendix A and Methods section. Cells from both the control and MeTC7-treated populations were imaged at 200× (20× objective, 10× ocular) magnification. Image analysis was executed by a single analyst to maintain sampling consistency. AML cells stained with CellTracker Green (Thermo Fisher, Cat# C7025) were counted manually. For area quantification, ImageJ (Fiji) software was used. A total of 400 randomly selected AML cells from each population were manually traced to ensure accuracy, and as such subsequent statistical analyses were executed using *n* = 400 for each respective population. The area of traced cell was calculated by software calibrated to image parameters. Cell viability dye intensity was determined semi-quantitatively, determining color expression via reciprocal intensity and analyzing the same 400 randomly selected cells of each population as described previously.

Efferocytosis assay: THP-1 cells seeded at 1 × 10^5^ density in a 24-well plate in RPMI + 10% FBS + 1% pen-strep were treated with MeTC7 (340 µL) or vehicle for 48 h. Neutrophils were isolated from human peripheral blood (under IRB approval) via Mono-Poly resolving medium according to the manufacturer’s instructions and incubated at −80 °C in FBS + 10% DMSO for a minimum of 24 h to induce apoptosis. Apoptotic neutrophils were washed with PBS and fluorescently labeled with 2 μM PKH26GL (Sigma-Aldrich, Saint Louis, MO, USA, # PKH26GL-1KT) according to the manufacturer’s instructions. End-stage neutrophils were then provided in excess (10:1) to plated THP-1 cells for 3 h. Cells were then imaged and collected for flow cytometry analysis. All samples were run on an LSRII flow cytometer using FSC, SSC, 355 nm (DAPI), and 535 nm (PKH26) (BD Biosciences, Franklin Lakes, NJ, USA). Analysis was performed using FlowJo version 10.7.1.

Syngeneic colorectal cancer model: MC38 cells (1 × 10^5^/mice) derived from murine colonic adenocarcinoma were injected intramuscularly in the left legs of female C57BL/6J. Mice were locally irradiated using a 3200 Curie-sealed ^137^Cesium source that operates at roughly 1.90 Gy/min, 7 days after tumor cell injection [26]. Jigs were constructed and designed to specifically treat the tumor-bearing leg with 15 Gy radiations. This source and the collimators used are calibrated periodically to ensure equal distribution of radiation. Standard calipers were used to measure tumor growth as described previously. Tumor-bearing mice were administered 25 mg/kg of MeTC7 or vehicle control (40% Hydroxypropyl-beta-cyclodextrin (Acros Organics, Geel, Belgium, # 1695740) +solutol HS15 (Sigma, San Francisco, CA, USA, # 42966) in sterile water) subcutaneously 1X/day starting 2 days before irradiation for the indicated duration of treatment.

U937 xenograft assay: U937 cells (5 million/mice) were mixed in 1:1 cold Matrigel:RPMI medium and injected subcutaneously in NSG mice. After a week, mice were treated with 10 mg/kg/IP (M-F), and tumors were measured in terms of length and width every 3 days.

Statistical analysis: The number of CFUs in the treatment versus control groups was analyzed by non-parametric *t*-test using GraphPad Prism 8.4.0 or earlier.

## 3. Results

VDR is overexpressed in AML subtypes and predicts poor prognosis: Patients with AML inv(16)/t(16;16), complex AML, or t(11q23)/MLL exhibited significantly elevated VDR mRNA expression than hematopoietic stem cells (HSC) controls (Figure 1A). Kaplan–Meier analysis showed that AML patients overexpressing VDR (*n* = 8) face increased mortalities than those with lower expression (*n* = 409) (*p* = 0.04) (Figure 1B). Western blot analysis of primary and relapsed AML cells (except patient 7) showed elevated VDR expression compared with normal bone marrow cells (Figure 1C). About half of the AML patient samples showed high PD-L1 expression (Figure 1C). Similarly, although not significant, PD-L1 mRNA overexpression indicated the trend for increased mortalities in AML patients (Appendix A).

VDR induces PD-L1 expression in AML and ovarian cancer cells: Calcitriol treatment induced upregulation of PD-L1 in primary AML (AML-1 and AML-2) cells (Figure 1D). Similarly, siRNA knockdown of VDR in MV411 (Figure 1E) and THP-1 (Figure 1F) cells, compared to scrambled siRNA controls, showed decreased PD-L1 expression. Original scans of the Western blots are presented in Appendix A.

VDR/vitamin D receptor response element (VDRE) interactions in PD-L1 promoter: ChIP assay (Appendix A) was performed following the method published by Dimitrov et al. [7] to determine VDR’s interaction with PD-L1 promoter. VDR/VDRE expression in the transcripts was captured by a VDR antibody in AML (Appendix A) and myelodysplastic syndrome (MDS) (Appendix A). In all these cell lines examined, the PCR products for the binding site could be visualized from those precipitated by the VDR antibody. Dimitrov et al. [7] showed vitamin D/VDR regulation on PD-L1 in THP-1 AML cells. We determined that similar to THP-1 cells, MV-411 and U937 also exhibit vitamin D/VDR upregulation on PD-L1. Importantly, (MDS), a precursor to AML, also showed vitamin D/VDR regulation of PD-L1. MDS are a rare group of bone marrow failure disorders, a portion of which converts into AML.

STAT3 inhibitor does not block calcitriol-induced PD-L1 expression in THP1 AML cells: In AML and MDS, STAT3 serves as a pro-leukemogenic transcription factor that is associated with poor prognosis and short disease-free survival [27]. STAT isoforms (1, 2, 3, and 5) are situated on the promoter of PD-L1 (Appendix A). We examined whether the pretreatment with Stattic, a characterized inhibitor of STAT3, could block calcitriol-induced PD-L1 expression. Stattic pretreatment for 6 h did not block calcitriol-induced PD-L1 upregulation; instead, increased PD-L1 expression was observed (Appendix A), indicating that vitamin D/VDR regulation of PD-L1 is Stat-3 independent.

MeTC7 is a specific VDR antagonist: The heterodimerization of RXR and VDR (RXR-VDR) and homodimerization of RXR (RXR-RXR) in an M2H demonstrated competitive inhibition by a VDR antagonist MeTC7 (Figure 2A) of only VDR-RXR heterodimers but not RXR-RXR homodimers, indicating the selective nature of the association of MeTC7 with VDR (Figure 2B). MeTC7 also demonstrated specific VDR inhibition when combined with 1,25D but did not compete nor bind with non-VDR receptors like RXR under any of the concentrations tested. Heterodimerization of VDR with transcriptional coactivators, SRC1 (Figure 2C) and DRIP205 (Figure 2D), was also effectively inhibited by MeTC7 at multiple concentrations. MeTC7 also exhibited selective competitive inhibition of VDR binding to the now-recognized alternative VDR agonist 3-keto lithocholate (3-K LCA, Figure 2E). This latter observation suggests that MeTC7 binds in the VDR ligand-binding pocket in such a way as to exclude two different natural VDR ligands (1,25D and 3-K LCA).

MeTC7 treatment reduced PD-L1 expression in AML cells: Next, we investigated the effect of MeTC7 on the expression of PD-L1 and PD-L2 in primary AML and MDS cells. Effects of MeTC7 on PD-L1 and PD-L2 expression were interrogated by immunoblotting. MeTC7 (100–500 nM) treatment for 48 h reduced the expression of PD-L1 in AML-1 (Figure 3A), MV-411 (Figure 3B), and THP1 (Figure 3C) cells. Compared to MV-411 cells, MeTC7 needed a 2.5-fold higher dose (250 nM) to inhibit PD-L1 in AML cells. Further, MeTC7 treatment did not inhibit PD-L2 expression in THP-1 (Figure 3D) and MV-411 (Figure 3E) cells. Original scans of the Western blots are presented in Appendix A.

MeTC7 treatment inhibited the growth of AML cells: Treatment with MeTC7 reduced the viability of THP-1, MOLM13, MV-411, and U937 cell lines (Figure 4A) and AML cells.

Collected from five patients (Figure 4A—middle) and three relapsed patients (Figure 4A—right). The phenotypic characteristics of patients-derived AML cells used in this study are shown in Figure 4B. Next, the effect of MeTC7 treatment on the clonogenic potential of AML cell lines was investigated. MeTC7 treatment reduced the size and number of colonies formed by THP-1 (Figure 4C—left) or U937 (Figure 4C—right and Figure 4D) in a dose-dependent manner. Images (20×) of the U937 colonies treated with vehicle compared to MeTC7 are shown (Figure 4D). MeTC7 treatment at 10 µM showed fewer colony units of primary AML cells, similar to Arac-C treatment, than vehicle (Figure 4E).

MeTC7 prevents AML adhered to MSCs: MSCs support the survival and proliferation of primary human AML cells [28]. Co-culture of AML-MSCs was shown to enhance the proliferation of AML cells, as well as chemoresistance [28]. We employed an AML-MSC co-culture method to investigate whether MeTC7 treatment could block adherence of primary AML cells with MSCs. MeTC7 treatment, under co-culture conditions with MSCs, significantly reduced cell counts, sizes, and viability of primary AML adhered with MSCs (Figure 5). MeTC7 suppressed AML cell viabilities selectively without affecting the viability of MSCs.

MeTC7 augments the efferocytotic efficiency of THP1 cells: Phagocytosis of dead and dying cells, also termed efferocytosis, is essential for tissue homeostasis and maintenance [29]. Defective efferocytosis underlies the causes of a growing list of inflammatory diseases [30]. Blockade of PD-1/PD-L1 in vivo has been shown to increase macrophage efferocytosis [31]. We examined if VDR/PD-L1 inhibition by MeTC7 could affect the efferocytic efficiency of THP-1 macrophages by measuring repeated efferocytic activity (Figure 6A). MeTC7 treatment increased the efferocytic efficiency of THP-1 cells (Figure 6B, right), while the total engulfment was not increased (Figure 6B, middle) after MeTC7 treatment.

MeTC7 inhibits radiotherapy (RT)-induced PD-L1 expression in vivo: MC38 colorectal cancer cell-based syngeneic animal model [26] in which VDR and PD-L1 overexpression co-occurs under radiation therapy was employed to estimate the efficacy of MeTC7. The treatment of MC38 colorectal cancer cell-derived tumors (Schema: Figure 7A) with MeTC7 growing in mice showed that MeTC7, in combination with RT, significantly reduced the surface expression of PD-L1 on tumor cells (Figure 7B). In addition, MeTC7 treatment in combination with RT increased CD8+T-cell infiltration (Figure 7C), and the percent of CD8+ T cells expressing CD69 and PD-1 activation markers increased compared to vehicle or RT + vehicle (Figure 7D,E).

## 4. Discussion

Our study provides a small-molecule MeTC7 that can inhibit PD-L1 expression in AML (Figure 3). Development of MeTC7 as the inhibitor of VDR/PD-L1 assumes importance in the context of calcitriol’s role in the upregulation of PD-L1 (Figure 8), enrichment of Tregs [32], suppressions of CD8+T cells [33], and stunting of NK cell growth [34] and T-cell activation [35], each of which can be exploited by tumors to evade immune detection. MeTC7 selectively reduces PD-L1 in AML cell lines, sparing PD-L2 (Figure 3D,E). Both PD-L1 and PD-L2 compete for binding with the PD-1 receptor with similar affinity. Lack of activity against PD-L2 is desirable because PD-L2 is associated with the functions of macrophages, DCs, mast cells, and vascular endothelial cells and plays critical roles in normal immunity. To validate the anti-PD-L1 effects of MeTC7 in vivo, we utilized a syngeneic colon cancer model in which radiation therapy (RT) induces overexpression of both VDR and PD-L1 on tumor cells. Although MeTC7 initially slowed the growth of AML xenograft in NSG mice (Appendix A) despite starting at higher average volume sizes, the RT-based model was considered more informative because of the convergence of VDR and PD-L1 and the presence of the intact immune system. In this model, MeTC7 reduced PD-L1 expression and increased lymphoid activation marker CD69+ on CD8+ T-cell populations in the tumors, consistent with PD-L1/PD-1 pathway blockade.

MeTC7 is both structurally and mechanistically different from BMS202 (and analogs), CA-170, and INCB086550, the other leading small molecules that were shown to target the PD-L1/PD-1 axis. BMS202 and INCB086550 block PD-L1/PD-1 interactions, mimicking the therapeutic actions of antibodies [18,20,21,22,23]. MeTC7 inhibits surface and intracellular PD-L1. Given the potential of MeTC7 in inhibiting PD-L1, it was considered important to validate that MeTC7 was free from any residual VDR agonistic effects. The literature described antagonists carrying partial VDR agonistic effects, which can overexpress PD-L1. M2H assay showed that MeTC7 not only demonstrated specific VDR inhibition when combined with calcitriol but also did not compete nor bind with non-VDR receptors like RXR under any of the concentrations tested (Figure 2). Heterodimerization of VDR with transcriptional coactivators SRC1 and DRIP205 was also effectively inhibited by MeTC7.

The rationale for targeting PD-L1 in AML stems from: (1) overexpression of PD-L1 in AML and MDS, the precursor of AML, and their association with poor prognosis (Appendix A) [36,37]; (2) PD-L1 enrichment in AML blasts at post-transplantation relapse; (3) PD-L1 orchestrated AML relapses after allogeneic hematopoietic cell transplantation (allo-HCT) [38] and (4) PD-L1 induced secondary resistance to hypomethylating agents (HMAs) in AML [39]. MeTC7 can also be effective in enhancing the response of HMAs to both AML and its precursor neoplasm, MDS since HMAs increase PD-L1 expression in MDS and AML. The activity of anti-PD-L1 antibodies among AML patients and those with relapsed disease was shown [40]. Among AML subtypes, inv(16)/t(16;16), t(11q23)/MLL, and AML complex, which exhibit VDR mRNA overexpression (Figure 1A) and can be targeted by MeTC7. VDR/PD-L1 inhibition by MeTC7 results in loss of viability and colony formation capabilities of primary and relapsed AML cells.

Further, our studies show that targeting the VDR/PD-L1 axis prevents AML cells’ attachment to MSCs (Figure 5). Interactions of AML/MSCs and MDS/MSC cells enable MDS and AML’s genesis, progression, chemoresistance, and relapse [41,42]. Further, MeTC7 treatment can enhance the phagocytotic activities [43] of macrophages (Figure 6). Defective efferocytosis is one of the underlying causes of malignancies [44,45]. Efferocytosis of apoptotic neutrophils by macrophages was shown to promote anti-inflammatory signaling, prevent neutrophil lysis, and dampening of immune responses [46]. Similarly, tumor intrinsic and intracellular PD-L1 signals enhance cancer cell survival and confer resistance to anti-PD-1/PD-L1 antibody therapies [47]. Silencing PD-L1 in MDA-MB-231 breast cancer cells increased apoptosis and enhanced susceptibility to doxorubicin in vitro and in vivo [48]. Similarly, CRISPR/Cas9 knockout of PD-L1 enhanced the sensitivity of human osteosarcoma cells to doxorubicin and paclitaxel and blocked the ability to form three-dimensional spheroids in vitro [49].

Potential clinical applications of this study and specifically of MeTC7 reside in the small-molecule immunotherapy of AML, ovarian cancer, and pancreatic cancer, the cell lines of which show vitamin D/VDR dependencies of PD-L1 expressions.

A major limitation of this study is the lack of an in vivo AML model that can recapitulate the vitamin D/VDR/PD-L1 axis, which would validate the above-shown outcomes of MeTC7 treatment in the AML tumor microenvironment.

## 5. Conclusions

Taken together, based on the inhibitions of AML cell viability, colonies formed and PD-L1 expression, and increased phagocytosis, we anticipate that MeTC7 can improve the survival in AML via inhibition of PD-L1. Inhibition of tumor intrinsic and intracellular PD-L1 by MeTC7 would prevent PD-L1 rebound on the tumor cell surface and block the emergence of resistance. In conclusion, MeTC7 presents as a promising small molecule to inhibit PD-L1 in vitro and in vivo.

## Figures and Tables

**Figure 1 cancers-15-03432-f001:**
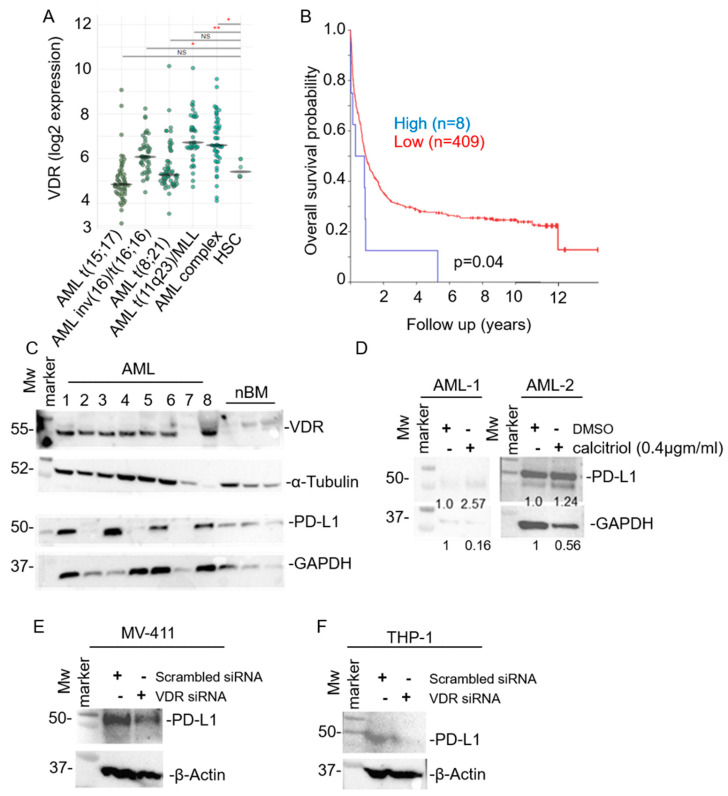
VDR is upregulated in AML subtypes and shows poor prognostication and upregulation of PD-L1 upon calcitriol stimulation. * *p* < 0.05; ** *p* < 0.01; (**A**) The VDR mRNA expression in AML patients and survival probability were analyzed using R2-Genomics Analysis and Visualization Platform (Bohlander-422-MAS5.0_u133a). (**B**) Overall survival in AML patients based on VDR expression (high: *n* = 8; low: *n* = 409) showed decreased survival. (**C**) Immunoblotting of VDR and PD-L1 in cells from randomly selected primary (*n* = 5) and relapsed (*n* = 2) AML patients, compared to normal bone marrow (nBM) (*n* = 3). (**D**) Immunoblotting of PD-L1 in AML-1 and AML-2 cells with calcitriol (400 nM) stimulation for 48 h. The densitometry analysis is shown below the bands. Immunoblotting of PD-L1 in MV-411 cells (**E**) and THP-1 cells (**F**) expressing scrambled siRNA or VDR siRNA. The uncropped blots are shown in Appendix A.

**Figure 2 cancers-15-03432-f002:**
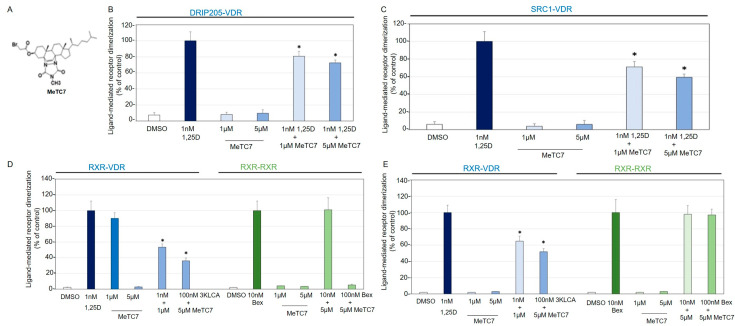
MeTC7 is a potent VDR antagonist. (**A**) Chemical structure of MeTC7. (**B**) MeTC7 demonstrates selective competitive inhibition. Heterodimerization of RXR-VDR in response to 1,25D and/or MeTC7, and homodimerization of RXR-RXR in response to bexarotene and/or MeTC7 in a mammalian 2-hybrid system (M2H). (**C**) VDR binding to receptor co-activator SRC1 in an M2H assay. (**D**) VDR binding to receptor co-activator DRIP205 in an M2H assay. (**E**) Selective competitive inhibition of VDR agonist 3-keto lithocholate by MeTC7. Results are plotted as ligand-mediated RXR-VDR heterodimerization (or RXR-RXR homodimerization) compared to the positive control (1,25D or bexarotene) set to 100%. The negative control is DMSO. Error bars represent standard deviations (* *p* < 0.05 versus 1,25D control). The data are a compilation of between six and eight independent assays, with each treatment group dosed in quadruplicate for each independent assay. The transcriptional activation of the reporter gene was measured in comparison to the reference compound 1,25D or bexarotene. Error bars indicate the standard deviation of the replicate experiments. Different shadings of blue in panels B-E represent the VDR-containing M2H systems, while different shadings of green represent the RXR-only M2H. White fill represents the DMSO negative (vehicle) control.

**Figure 3 cancers-15-03432-f003:**
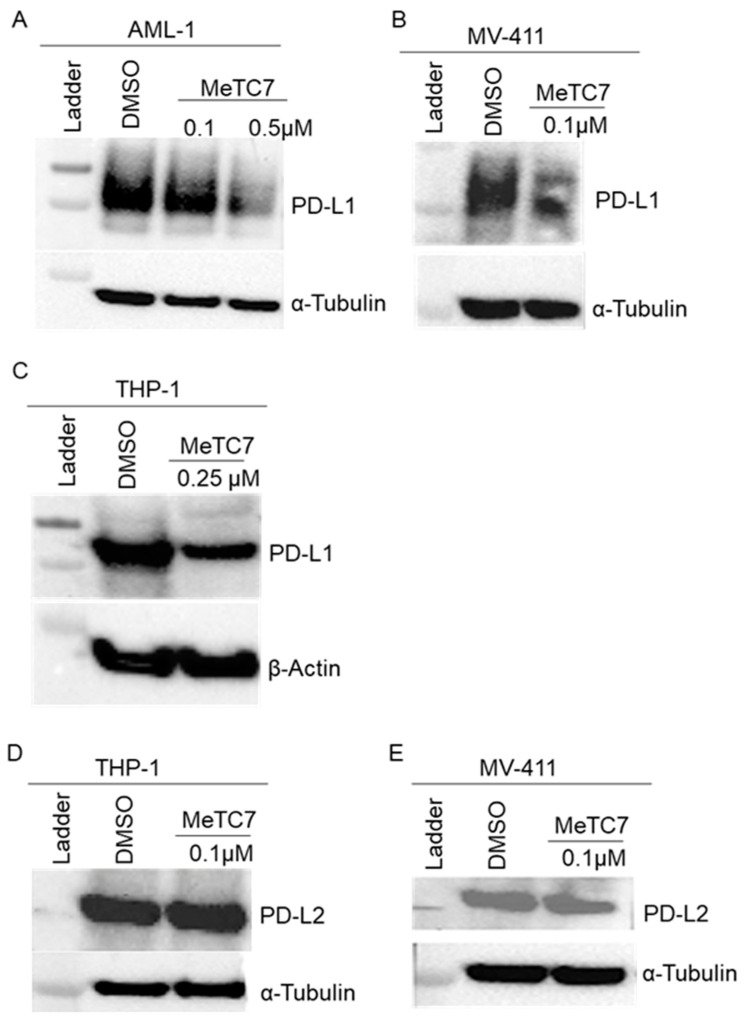
MeTC7 selectively inhibits PD-L1 expression in AML cells: MeTC7 treatment reduced PD-L1 expression in AML-1 primary cells (**A**), MV-411 (**B**), and THP-1 (**C**) cells during 48 h of indicated dose treatment. MeTC7 treatment at the indicated doses did not alter the expression of PD-L2 in THP-1 and MV-411 cells (**D**,**E**). The uncropped blots are shown in Appendix A.

**Figure 4 cancers-15-03432-f004:**
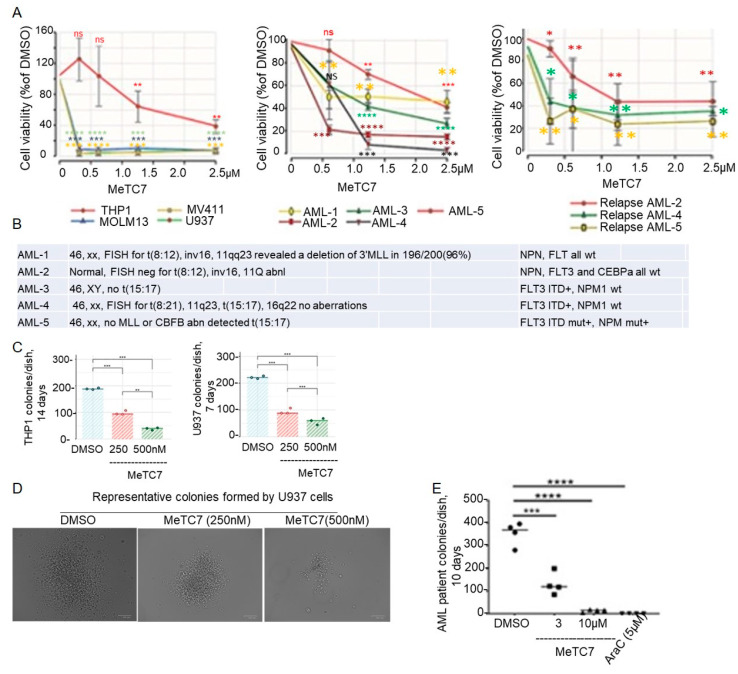
MeTC7 inhibited the proliferation of AML cells dose dependently. MV411, THP1, U937, and MOLM13 cell lines (**A**—**left**),five randomly selected primary human AML cells (**A**—**middle**) or three relapsed AML cells (**A**—**right**) were seeded and treated with DMSO or MeTC7 in complete RPMI-1640 medium for 40 h. MTS (Promega, Celltiter96Aqueous one solution, #G3580) (10 µL) was added to each well and to those without cells (blank) and incubated for an additional 4–6 h. The optical density of the wells was recorded using the BioRad iMark microplate reader at 490 nM wavelength. The percentage viability of MeTC7-treated cells was calculated relative to DMSO-treated cells. Data as mean ± SEM are shown. (**B**) Table: Cytogenetic characteristics of five human primary AML cells used in Figure 5B are shown. MeTC7 treatment inhibited colonies formed by THP-1 (**C**—**left**) and U937 (**C**—**right**) cells. THP1 and U937 cells were added to MethoCult H4435 Enriched (Stem Cell Technologies, #: 04435) to achieve a concentration of 2000 cells/mL. DMSO (<0.2%) or MeTC7 (250 and 500 nM) was added. Each condition was performed in triplicate. The plates were incubated at 37 °C with 5% CO_2_ in a humidified incubator. Colonies were counted on day 7 (U937 cells) and day 14 (THP1 cells). One-way ANOVA and Tukey’s multiple comparison tests of the number of colonies formed in control versus treatment groups were performed using GraphPrism-8. * *p* < 0.05; ** *p* < 0.005, *** *p* < 0.0005, **** *p* < 0.00005. (**D**) Representative light contrast images of the U937 cell colonies treated with vehicle or MeTC7 (250 and 500 nM) on day 10 of the experiment: (**C**—**right**) growing into MethoCult H4435 enriched (Stem Cell Technologies, Canada, # 04435) are shown. (**E**) MeTC7 treatment reduced the colony units formed by a primary AML cell in MethoCult H4435 enriched dose dependently. Ara-C was used as a positive control. Details of the experimental procedures are described in the Appendix A section.

**Figure 5 cancers-15-03432-f005:**
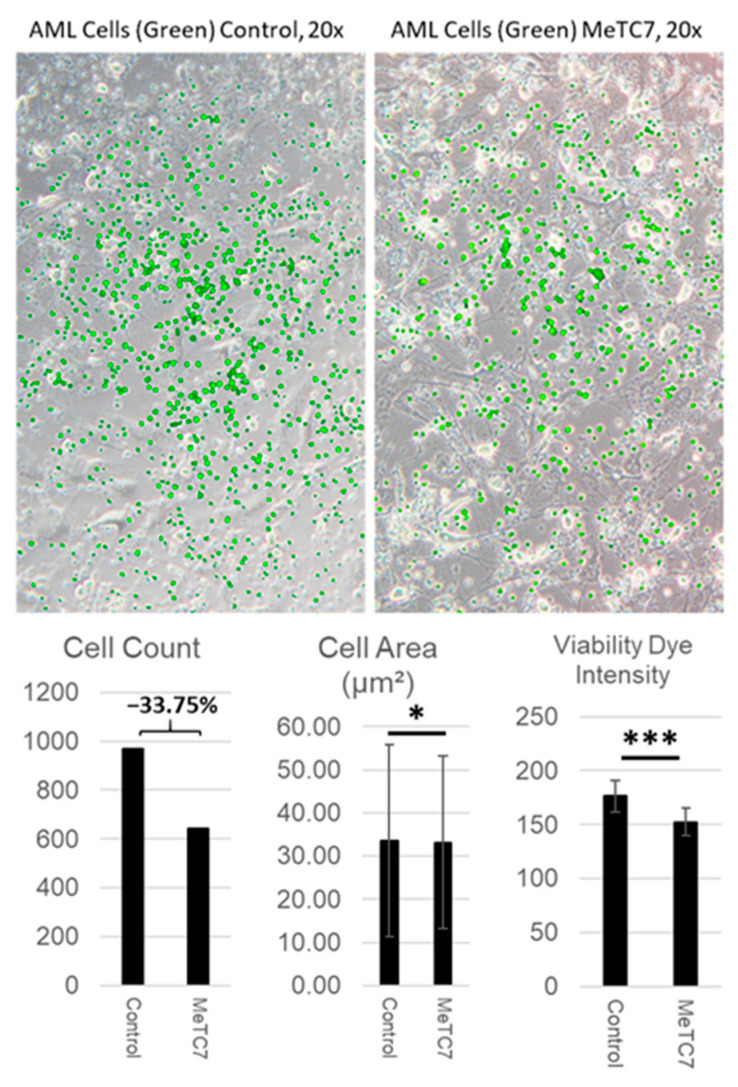
Effect of MeTC7 on the attachment of primary AML cells with MSCs. MSCs were co-cultured with pre-stained CellTracker Green for 24 h. Control (DMSO) and MeTC7 (300 nM) treatment occurred by the addition of media-containing treatment into an equal volume of co-culture-containing media. At 24 and 48 h after treatment, cells were imaged using an Olympus CKX-41 inverted microscope equipped with epifluorescence illumination and the U-FF fluorescence filter kit (excitation 467–498 nm). Image acquisition used Olympus cellSens imaging software version 2.3. Background subtracted fluorescence images were merged with brightfield images using Adobe Photoshop, release 23.2.2. Annotation for * *p* < 0.05 and *** *p* < 0.0005. For area quantification, ImageJ (Fiji) software (ImageJ 1.53t, Java 1.8.0_322 (64-bit) was used. Randomly selected AML cells (*n* = 400) from each population were manually traced. Statistical analyses were performed using *n* = 400 respective cell populations. Area of traced cell calculated by software was calibrated to image parameters. Cell viability (dye intensity) was determined semi-quantitatively utilizing the color expression via the reciprocal intensity of images using ImageJ (ImageJ 1.53t, Java 1.8.0_322 (64-bit).

**Figure 6 cancers-15-03432-f006:**
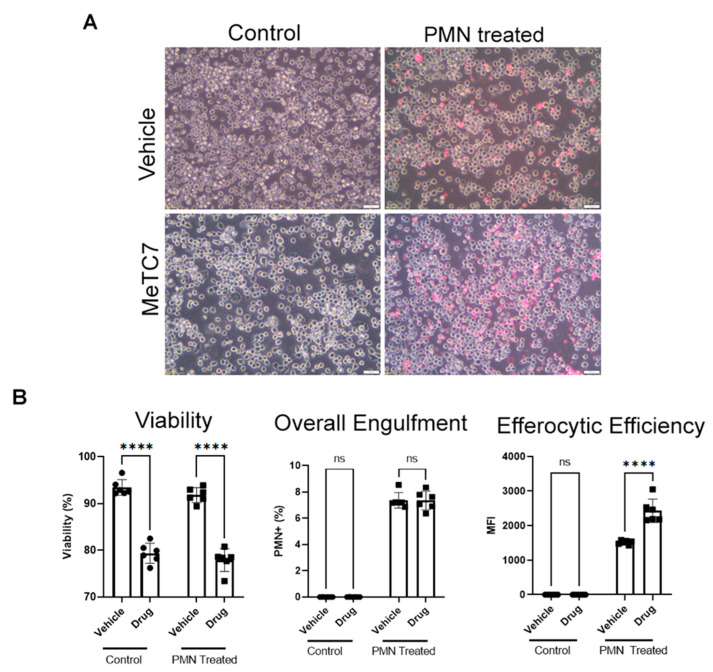
MeTC7 enhances the phagocytotic activity of THP1 cells. THP1 cells were treated with MeTC7 (340 nM) or vehicle for 48 h. Neutrophils (PMNs), isolated from human peripheral blood, were incubated at −80 °C in FBS+ 10% DMSO for 24 h to induce apoptosis. Apoptotic neutrophils were washed with PBS and fluorescently labeled with 2 μM PKH26GL. End-stage neutrophils were provided in excess (10:1) to plated THP1 cells for 3 h. Cells were imaged and collected for flow cytometry analysis. All samples were run on an LSRII flow cytometer using FSC, SSC, 355 nm (DAPI), and 535 nm (PKH26). Analysis was performed using FlowJo version 10.7.1. Annotation for ns: not significant, **** *p* < 0.0001. Images were taken at 20×. Scale bar = 50 µM.

**Figure 7 cancers-15-03432-f007:**
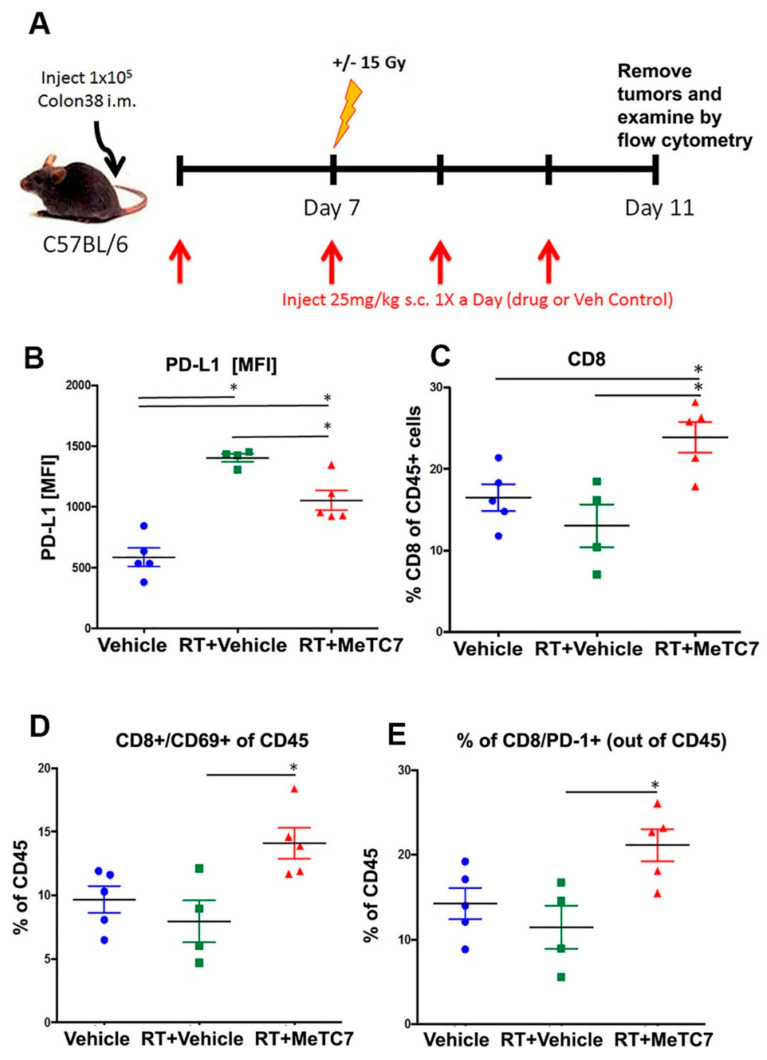
MeTC7 reduces PD-L1 expression in an RT-inducible PD-L1overexpression syngeneic model. (**A**): Schema of determination of activity against PD-L1 activated by RT in an orthotopic model of MC38 colorectal cancer in BL7 mice. MeTC7, in combination with RT-abrogated RT-induced PD-L1 activation in CD45^−^ tumor cells post-RT (**B**); (**C**), increased CD8+T-cell infiltration in tumors significantly compared to vehicle or RT + vehicle. (**D**,**E**): MeTC7 increased the %of cells expressing CD69 (**D**) and PD-1 (**E**) compared to vehicle and vehicle + RT. The details of the antibodies used are listed in Appendix A. Annotation for * *p* < 0.05.

**Figure 8 cancers-15-03432-f008:**
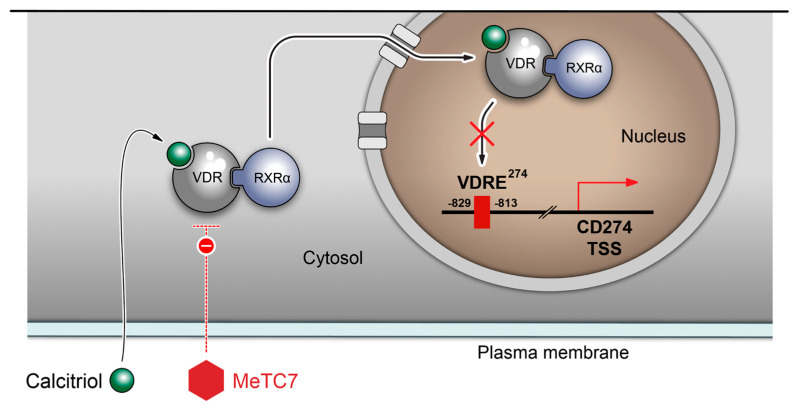
Proposed signaling effects of MeTC7 against VDR/PD-L1 in cancer cells.

## Data Availability

Raw data will be provided upon reasonable request.

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
