# Peer review of "Vitamin D Receptor Antagonist MeTC7 Inhibits PD-L1"

_cancers, 2023, doi:10.3390/cancers15133432_

Round 1

Reviewer 1 Report

Dear Authors,

This is a complex study addressing the issue of a particular vitamin D receptor antagonist, namely the molecule MeTC7, in order to inhibit programmed cell death ligand 1 (PD-L1) which plays an important immune role in human body specifically addressing malignancy domain. A meticulous presentation of results is displayed while the topic itself opens up to

Here are my observations:

1.       No need for point after the authors list

2.       Please use the same size font when listing the number of affiliations

3.       The sign for correspondent author should be the same

4.       Title/Abstract. PD-L1 should be explained when first used (for it stands for).

5.       There are unnecessary spaces before “Furthermore..: (line 102), “Examples.. (line 91)

6.       At the end of Introduction the aim of the study should be clearly mentioned (the importance should be discussed after Results)

7.       The first section at Methods should be the study design

8.       Discussion – At the parts when you mention “data not shown” I suggest to either introduce some supplementary materials, either provide the data as a figure/capture, either cite a reference

9.       Discussion. Please introduce a brief comment on further potential clinical applications of this study

10.   The last section at Discussion should provide the limits of the current study

11.   “Conclusion” section should introduce the data that are already presented within the main text; no reference should be mentioned at this section

Thank you,

Author Response

Please see the attachment. Thank you for your great comments and suggestions.

Reviewer 2 Report

The group studied the potential drug that targets the vitamin D receptor (VDR) to suppress PD-L1 expression in which the suppression could augment the immune checkpoint inhibitor treatment against blood cancer and solid tumors. MeTC7 binds to VDR to prevent the protein from undergoing heterodimerization with coactivators and calcitriol. Here are my comments for this manuscript:

1)      Figure 1B: Kaplan Meier survival analysis showed marginal statistically insignificant. Please provide more convincing datasets to stress high VDR expression is correlated with poor prognosis in AML patients.

2)      Figure 1C-F: the quality of western blotting results are poor. Please provide results with equal amount of α-tubulin, GAPDH, and β-actin for C and D.

3)      Figure 1D: since the group managed to quantify the intensity, I think it is better to perform appropriate quantitative analysis with significance values (p-values).

4)      Supplementary figures (Figure S2-S4) as mentioned from Pg 7_line 219-236 are not available. I only manage to download the original images for blots/gels.  

5)      “Compared to MV-411, MeTC7 needed 2.5- fold higher dose (250nM) than MV-411 cells to inhibit PD-L1.” It is unclear to me. Did you mean AML-1 vs MV-411?

6)      Figure 4A-C: no statistical analysis.

7)      Figure 4G: please provide images with higher resolution.

8)      Figure 5: no SD bars and sample size.

9)      It is quite an abrupt shift from AML to CRC. I understand that the group wants to replace another cancer cell line with similar correlation (high VDR expression is correlated with high PD-L1 expression). However, intrinsic cancer biology is different between AML and CRC. I suggest the group performs IHC to demonstrate MeTC7 promoted CD8+ T cells to interact with cancer cells. 

Regarding Cartoon-1, the group emphasized that VDR only interacts with Calcitriol to regulate PD-L1 expression level. I would suggest the group amends it because the findings here imply that VDR could regulate PD-L1 expression without Calcitriol. For example: 

a)    The group demonstrated that MeTC7 treatment suppressed the PD-L1 expression in AML-1, MV-411, and THP-1 cell lines. Did the group treat both DMSO group and MeTC7 group with calcitriol? If the group didn’t, these data could imply VDR could regulate PD-L1 expression independent of calcitriol given that MeTC7 is VDR-specific antagonist.

b)    In the animal experiment, did radiation also increase calcitriol level in mice? If not, the findings could suggest that VDR mediates PD-L1 expression independent of calcitriol.

The group should elaborate any possibilities that VDR promotes PD-L1 expression without calcitriol.

Author Response

please see the attachment. Thank you for your thought-provoking questions and comments, which made our manuscript so much better.